# Position: Benchmarks for Vision–Language Models in Urban Perception Should Be Reliability-Aware and Negotiated

**Rashid Mushkani** [1] [2]

## Abstract

Vision–language models (VLMs) are increasingly used to generate structured descriptions of street-level imagery for tasks such as streetscape auditing, mapping, and public consultation. These uses combine observable attributes with appraisal categories, and the human targets are often distributions of judgments with disagreement and explicit non-response. This position paper argues that benchmarking VLMs for urban perception should treat disagreement and abstention as measurement outcomes, report inter-annotator reliability alongside model alignment, and treat the label space and scoring policy as negotiable artifacts when outputs are intended to inform urban governance. We ground the argument in a benchmark of 100 Montreal street scenes annotated along 30 dimensions by 12 participants from seven community organizations, and in a deterministic zero-shot evaluation of seven VLMs. Across dimensions, model agreement with human consensus co-varies with dimension-level human reliability, and for the appraisal dimension *Overall Impression* models and annotators exhibit distributional mismatch including different rates of `Not applicable`. We close with actions for benchmark creators, model developers, and institutions to make uncertainty and benchmark assumptions visible in evaluation reports.

## 1. Introduction

Urban governance and planning use descriptions of place that combine physical form with perception. Urban design scholarship operationalized legibility and imageability (Lynch, 1960), while urban computing developed data-driven accounts of city life (Zheng et al., 2014). Perception has been operationalized at scale through crowd labeling and pairwise comparisons of street imagery (Salesses et al., 2013), including datasets that support models approximating aggregate judgments (Dubey et al., 2016). In parallel, computer vision on street-level and satellite imagery has linked visual cues to socioeconomic patterns, urban form, and mobility (Fan et al., 2023; Jean et al., 2016; Yeh et al., 2020; Tatem, 2017).

Recent VLMs can be prompted for structured image analysis without task-specific training (OpenAI, 2023; Li et al., 2023; Dai et al., 2023). Their use in urban workflows introduces an evaluation problem that differs from many benchmarks where the target is treated as a stable label. Urban perception tasks often include appraisal categories (e.g., comfort, safety, inclusivity) whose interpretation varies by context and experience. Existing evaluations of spatial and geographic knowledge in foundation models document capabilities and biases (Gurnee & Tegmark, 2024; Roberts et al., 2023; Manvi et al., 2024), but they do not settle how to interpret model scores when the target itself contains disagreement, abstention, and plurality.

**Position: Benchmarks and evaluations of VLMs for urban perception should treat annotator disagreement and abstention as informative signals, report inter-annotator reliability alongside model scores, and be negotiated with affected communities before model outputs are used to justify urban decisions.**

Although urban perception is our motivating case, the core issue is a general benchmarking problem whenever the target label space is subjective or governance-laden: collapsing pluralistic judgments (and abstentions) into a single "ground truth" and scoring with point estimates can hide systematic disagreement and make accuracy contingent on the labeling population and uncertainty semantics (Sap et al., 2022; Irvin et al., 2019). The same failure mode is documented in (i) content moderation/toxicity detection, where toxicity perceptions vary with annotator identities and beliefs and deployed systems inherit particular perspectives (Sap et al., 2022); (ii) medical imaging, where CheXpert introduces explicit uncertainty labels and multi-radiologist reference standards because radiograph interpretation contains

[1]Université de Montréal, Montréal, Québec, Canada [2]Mila – Québec AI Institute, Montréal, Québec, Canada. Correspondence to: Rashid Mushkani <rashidmushkani@gmail.com>.

*Proceedings of the $43^{rd}$ International Conference on Machine Learning*, Seoul, South Korea. PMLR 306, 2026. Copyright 2026 by the author(s).

irreducible uncertainty (Irvin et al., 2019); and (iii) predictive policing, where models trained on police-recorded "discovered" crime can reinforce biased allocation through feedback loops, effectively predicting future policing patterns rather than underlying crime incidence (Lum & Isaac, 2016; Ensign et al., 2018). The ML community should care because these domains already use ML outputs to support institutional decisions, so benchmark design choices and documentation practices directly shape what model "progress" means and what harms are incentivized (Sap et al., 2022; Irvin et al., 2019; Lum & Isaac, 2016; Gebru et al., 2018).

The paper develops this position in two ways. First, it argues that benchmark scores for appraisal targets are conditional on measurement properties of the target, including the stability of human judgments and the semantics of abstention. Second, it provides an empirical anchor: a community-annotated benchmark of Montreal street scenes that makes disagreement and abstention observable, evaluates multiple VLMs under a fixed prompt contract, and reports model alignment together with reliability. The empirical evidence addresses three questions about this benchmark: how alignment varies between more observable dimensions and appraisal dimensions; how dimension-level reliability relates to dimension-level model alignment; and how alignment differs between photographs and photorealistic synthetic scenes. The remainder of the paper connects these observations to benchmark governance and to procedures for revising contested label spaces through negotiation.

## 2. Position and Scope

The position concerns evaluation practice and benchmark governance rather than a specific model family. It applies to domains where model outputs are likely to be inputs to institutional decisions and where the label space contains value-laden or context-dependent categories. Urban perception provides a concrete case because categories such as comfort, accessibility, and safety are used in planning discourse and are known to vary with social experience and local norms. In such settings, disagreement can be treated as annotation error, as plurality, or as evidence that a category boundary is not stable. When a benchmark collapses disagreement into a single label without reporting reliability, the resulting score can be interpreted as a property of the model even when it also reflects the properties of the labeling process.

Reliability-aware benchmarking has a descriptive component and an interpretive component. The descriptive component is that benchmarks report stability of judgments per dimension and disclose how judgments were collected, including language, recruitment, overlap structure, and the role of abstention. Reliability measures differ in assumptions and rater structure; Cohen's $\kappa$ and Fleiss' $\kappa$ target different rater regimes (Cohen, 1960; Fleiss, 1971). The interpretive component is that model alignment scores are presented as conditional on these measurement properties. In particular, for appraisal categories, low reliability indicates that a single summary label is a low-resolution representation of the target, and the score should be interpreted accordingly. This component is falsifiable: if, across perception benchmarks, reporting reliability and abstention rates does not change model comparisons, does not alter conclusions about deployment risk, and does not affect benchmark revision decisions, then the marginal value of this reporting requirement would be limited.

Negotiation is the third component of the position. Benchmark construction selects categories, definitions, translation mappings, and scoring rules that define what counts as alignment. In urban settings these choices interact with contestation and accountability mechanisms. The position therefore treats benchmarks as artifacts that can be revised through procedures that include affected communities. Negotiation does not imply that each deployment requires a new benchmark. It implies that benchmark assumptions are open to contestation, that revision processes exist, and that benchmark results are not presented as universal measurements detached from the process that produced the targets. Work on pluralistic targets, co-production lifecycles, and governance mechanisms in urban AI offers concrete structures for this claim (Mushkani et al., 2025b). In a related framing, negotiative alignment treats disagreement as a signal that can motivate revision of specifications rather than post hoc reconciliation.

## 3. Related Work

**Urban perception as a benchmark target.** Pairwise comparisons have been used to map perceived urban qualities at scale (Salesses et al., 2013), and datasets such as Place Pulse 2.0 supported learning models that approximate crowd judgments (Dubey et al., 2016). Street-view imagery has supported tasks that connect visual cues to urban indicators (Fan et al., 2023), while satellite imagery has been used for socioeconomic estimation and spatial demography (Jean et al., 2016; Yeh et al., 2020; Tatem, 2017). Urban vision–language work extends contrastive learning and pretraining to urban tasks, including UrbanCLIP and UrbanVLP (Yan et al., 2024; Hao et al., 2025). These lines of work increase the need for evaluation practices that clarify what constitutes alignment when labels are appraisals or when labels are heterogeneous across groups.

**Vision–language models, evaluation tooling, and spatial reasoning.** VLMs integrate vision encoders with language models (OpenAI, 2023; Li et al., 2023; Dai et al., 2023; Liu et al., 2024; Pichai & Hassabis, 2023). Evaluation toolk-

its emphasize standardized harnesses across models and tasks (Duan et al., 2024). Separate work probes spatial and geographic capabilities and biases in foundation models (Gurnee & Tegmark, 2024; Roberts et al., 2023; Mai et al., 2024; Bhandari et al., 2023; Manvi et al., 2024) and in embodied or street-view navigation settings (Chen et al., 2019; Mirowski et al., 2018; Schumann et al., 2024). This paper focuses on a distinct evaluation problem: alignment with human judgments when the target is pluralistic, abstentions occur, and downstream use requires that uncertainty and contestation remain visible.

# 4. Dataset, Participants, and Annotation Protocol

The benchmark serves as an empirical anchor for the position. It is not presented as a representative sample of urban perception and it is not a substitute for larger datasets. The design goal is to make observable how alignment depends on the reliability of judgments and on benchmark policies for aggregation and abstention.

**Image panels and sources.** We curated 100 street-level scenes in Montreal. Images are organized into ten panels ($p1\ldots p10$), each containing ten scenes. Panels $p1$–$p5$ consist of photorealistic synthetic renders, while $p6$–$p10$ contain photographs (50 images per source type). Synthetic scenes were reviewed and scenes with generation artifacts that interfered with interpretation were removed. Across panels we aimed to include variation in location type, time of day, vegetation presence, and crowding level.

**Participants and recruitment.** Twelve Montreal-based participants from seven community organizations annotated the images. Participants were recruited via partner organizations and compensated for their time. Self-identification for context was optional. Figure 1 summarizes self-reported context categories.[1] Ethics review was conducted and approved prior to recruitment and data collection.

**Dimensions.** The schema comprises 30 dimensions spanning setting, human presence and activity, built form and aesthetics, and subjective impressions. Each dimension has a concise definition and a fixed label set, distinguishing single-choice from multi-label items. Appraisal dimensions are included because they appear in planning discourse, but they are treated as pluralistic judgments rather than measurements. To reduce identity inference, *Demographic Diversity* records observed cues (e.g., mobility aids) rather than inferred identities.

---

[1]Categories were self-reported and optional; no individual-level demographics are released.

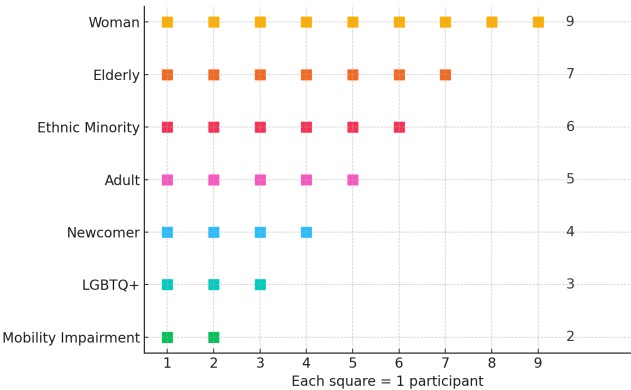

*Figure 1.* Self-reported participant context (counts). Categories are not mutually exclusive and reflect intersectional identities; participants could select multiple identity markers, so counts exceed the number of participants.

**Collection.** Each image received between one and three independent annotations, resulting in 230 completed forms. A shared subset ensured overlap across participants for reliability analysis. Annotation was conducted in French.

**Normalization and consensus.** We constructed a deterministic French-to-English mapping for each possible response string, covering synonyms and capitalization. For multi-label dimensions, we aggregated selections per image and formed a hard consensus set by retaining options selected by at least half of annotators ($\geq 50\%$). For single-choice dimensions, we used majority vote; exact ties were flagged *unsure* and excluded from accuracy calculations. The consensus target is used for scoring but is not interpreted as a latent ground truth.

Several dimensions include explicit abstentions (`Not applicable` and, for *Overall Impression*, `Cannot judge`). We treat these as non-response labels; by contrast, labels of the form `No X` denote observed absence and are scored as ordinary labels. For single-choice items, images whose consensus label is an abstention are excluded when computing accuracy. For multi-label items, we remove abstentions from both human and model sets and treat cases where both sets are then empty as missing, to avoid counting mutual non-response as overlap. Appendix E describes an alternative policy that counts abstentions as regular labels; the harness exposes both policies for sensitivity checks.

# 5. Models and Zero-shot Protocol

**Evaluated models.** We evaluate seven VLMs: `claude-sonnet`, `openai-o4-mini`, `gpt-4.1`, `gemini-2.5-pro`, `grok-2-vision`, `qwen2.5-vl`, and `llama-4-maverick`. All models were queried through the same script that embeds local images as base-64

data URIs and requests a single-line CSV response. For determinism we set `temperature=0`, `top_p=1`, and we log model version strings and run dates. This protocol mirrors a deployment mode in which a general-purpose model is prompted for a structured description without task-specific finetuning.

**Prompting and parsing.** The prompt enumerates the 30 dimensions in a fixed order with short definitions. It asks the model to choose one label for single-choice items and any number of labels for multi-label items. The script enforces a strict CSV format and retries on parse failures. The parser is rule-based and deterministic: it tokenizes the model CSV, trims whitespace, normalizes spelling, and maps tokens to codebook labels. The pipeline yields complete per-image, per-dimension model responses for all models; non-conforming replies are captured in a `Comments` column and excluded from scoring. The prompt contract is reported in Appendix F to make explicit how prompt design interacts with benchmarking.

## 6. Evaluation Methodology

**Scoring.** For single-choice dimensions we compute accuracy against the human majority label, excluding *unsure* items and items whose consensus label is an abstention (Section 4). For multi-label dimensions we compute the Jaccard index between the model-selected set and the human consensus set; abstentions are removed and cases where both sets are then empty are treated as missing. We average scores per dimension and report macro-averages across the 30 dimensions; denominators vary by dimension due to abstentions and ties.

**Human reliability.** Inter-annotator reliability is computed per dimension. We use Krippendorff's $\alpha$ with nominal distance; for multi-label items, each annotator set is treated as a nominal category, so $\alpha$ reflects exact-set agreement. Reliability is reported as descriptive metadata for interpretation rather than as a ranking signal.

**Additional analyses.** We analyze the relationship between human reliability and average model performance across dimensions; we compare performance across subsets of dimensions that are closer to observables versus subsets that involve appraisal; and we visualize distributional mismatch for the *Overall Impression* dimension. Synthetic-versus-real gaps are computed by stratifying images by source. For exploratory multiple comparisons we control the false discovery rate with the Benjamini–Hochberg procedure (Benjamini & Hochberg, 1995).

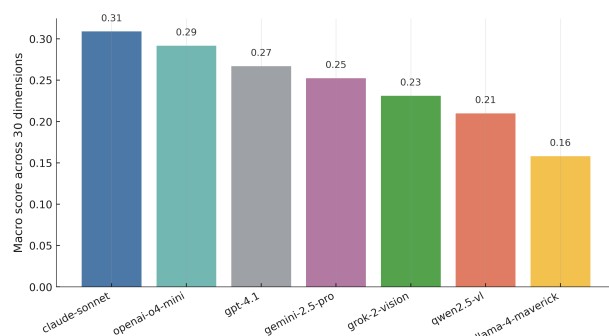

*Figure 2.* Overall agreement with human consensus by model. Macro-averaged accuracy (single-choice) and Jaccard (multi-label) across 30 dimensions.

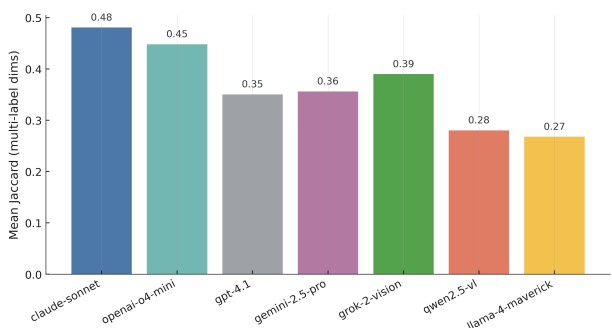

*Figure 3.* Mean Jaccard overlap between model selections and the human consensus for the multi-label dimensions.

## 7. Results

**Overall alignment summary.** Figure 2 reports a macro score (accuracy for single-choice items and Jaccard for multi-label items) as an interpretive summary rather than a leaderboard. Under the fixed zero-shot protocol, scores range from 0.16 to 0.31. To isolate multi-label behavior, Figure 3 reports mean Jaccard overlap with the human consensus on multi-label dimensions.

**Dimension-level variation and observability.** Figure 4 ranks dimensions by average model score. Dimensions referring to visible structure or presence occupy the upper part of the ranking, while dimensions with lower prevalence or with evidence that is difficult to infer from a single image occupy the lower part.

**Agreement structure across models and dimensions.** Figure 5 reports a heatmap of agreement with human consensus by dimension and model. The cross-dimension structure is similar across the evaluated models, which is consistent with the view that the schema and scoring policy define a mixture of dimensions with different measurement properties under a fixed prompt contract.

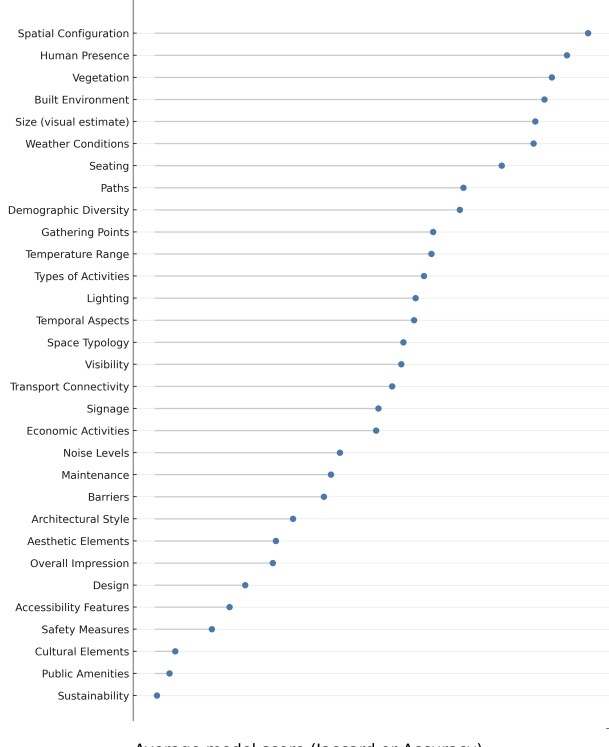

*Figure 4.* Difficulty by dimension. Each point is the mean model score for the dimension (accuracy or Jaccard depending on the item type).

**Human reliability and model performance.** Figure 6 reports Krippendorff's $\alpha$ by dimension. Figure 7 plots average model score against inter-annotator reliability. In this benchmark, the relationship is positive: dimensions with higher human agreement tend to have higher model agreement. Dimensions that deviate from this trend motivate targeted inquiry. For example, comparatively stable human agreement with lower model agreement is consistent with recoverable visual evidence that is not captured under the prompt contract or by the model. Conversely, higher model agreement with lower human agreement is consistent with model priors that reduce variance relative to the annotator distribution.

**Appraisal dimensions and observable dimensions.** We partition the schema into a subset closer to visible attributes and a subset involving appraisal. Figure 8 shows that most models attain higher macro scores on the subset closer to visible attributes than on the appraisal subset. This gap is relevant for deployments that use appraisal outputs as if they were observational measurements.

**Distributional mismatch on subjective appraisals.** Figure 9 compares the distribution of labels selected for *Overall Impression* across annotators and models. Several

models select `Not applicable` at higher rates than annotators and select *Accessible* at lower rates. This pattern is consistent with the prompt contract, which uses `Not applicable` for uncertainty, while annotators distinguish `Cannot judge` from `Not applicable`. Distributional reporting makes differences in priors and abstention semantics visible even when pointwise agreement is limited.

**Image source effects.** Figure 10 in Appendix C reports agreement stratified by photographic versus synthetic sources. In this benchmark, each model attains higher agreement on photographs than on synthetic renders. This gap supports reporting source metadata when synthetic images are used for data augmentation or evaluation.

## 8. Discussion

The empirical anchor is consistent with the position that evaluation practices should make disagreement, reliability, and abstention visible. The implications are interpretive and procedural.

First, the dimension-level variation in Figure 4 indicates that a macro score aggregates across targets that differ in relation to pixel evidence and differ in stability of judgment. For observables, low agreement is consistent with limitations in model recognition or in prompt specification. For appraisal categories, low agreement is consistent with unstable category boundaries, underspecified definitions, or missing contextual evidence. Reliability-aware benchmarking separates these interpretations by reporting stability measures next to alignment measures and by representing disagreement as a property of the target rather than as a residual to be removed.

Second, the positive relationship between reliability and model alignment in Figure 7 indicates that model comparisons can depend on the composition of the benchmark label space. When a benchmark includes dimensions with low reliability, model differences can reflect different priors for ambiguous cases rather than differences in recoverable visual evidence. This observation does not imply that low-reliability dimensions should be excluded. It implies that they require reporting formats that do not collapse disagreement into a single score. In the benchmark, distributional reporting on *Overall Impression* shows mismatches that are not summarized by accuracy.

Third, abstention is not only missingness but also a behavior that can differ between humans and models. In participatory settings, abstention can be interpreted as refusal, as insufficient evidence, or as a consequence of prompt semantics. Benchmarking that treats `Not applicable` as a regular label can inflate agreement, while benchmarking that ex-

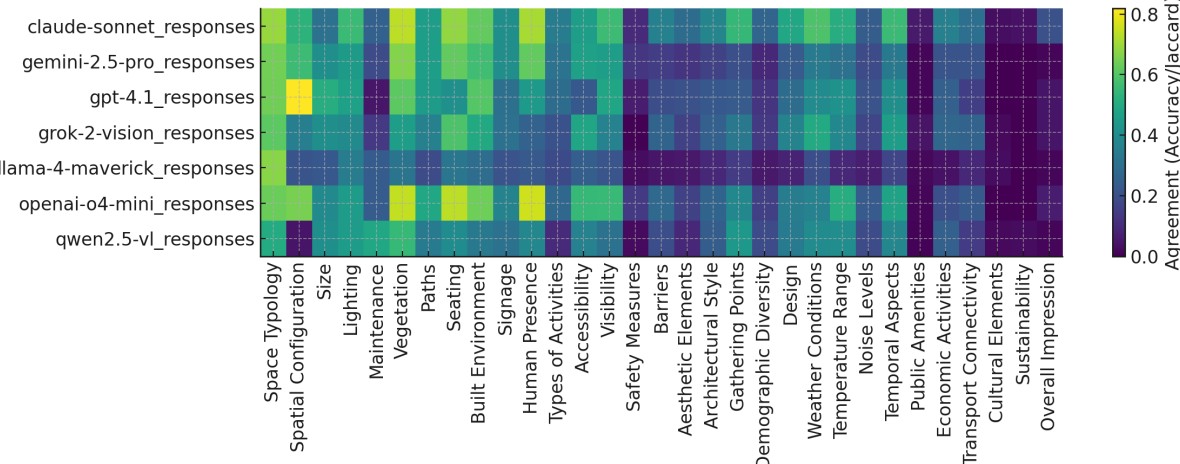

*Figure 5.* Agreement by dimension and model. Warmer colors indicate higher agreement with human consensus (accuracy for single-choice, Jaccard for multi-label).

cludes abstentions without documentation can obscure how often a model declines to provide an appraisal. Reporting abstention rates alongside alignment therefore changes what a score implies about the model and about the target.

Fourth, the source effect between photographs and synthetic scenes indicates that domain shift in urban perception is not limited to geography. Synthetic generation pipelines can alter cues that affect both human labeling and model inference, including texture, lighting, and composition. This supports stratified reporting and disclosure of the role of synthetic data when benchmarks are used to justify deployment.

These interpretive points connect to benchmark governance. Benchmarks define categories and scoring rules that shape model development incentives and that may be imported into institutional decision processes. Treating benchmarks as negotiable artifacts provides a mechanism for revising contested categories while keeping changes legible through versioning. In participatory urban AI, negotiation and specification revision are treated as governance mechanisms for pluralistic targets (Mushkani et al., 2025b). Operationally, a negotiated revision yields a new version of the label space, prompt contract, and abstention policy; re-annotation can test whether reliability, abstention rates, and model alignment shift.

## 9. Benchmark Negotiation and Versioning

Negotiation in this paper refers to a structured procedure for revising benchmark specifications when categories, definitions, or scoring policies are contested, or when the intended decision context changes. The procedure is distinct from post hoc interpretation because it produces an explicit specification artifact that can be inspected, compared across ver-

sions, and used to re-run evaluation. For urban perception benchmarks, the specification includes the label ontology, the semantics of abstention, language and normalization mappings, and the prompt contract that mediates between images and structured outputs.

A negotiated benchmark can be implemented without abandoning comparability by treating revisions as versioned task specifications. Under this framing, comparability is achieved by reporting results across versions and documenting the differences between versions, rather than by assuming that a single benchmark is context-free. Versioning also enables disagreement to be represented as structured alternatives, for example by maintaining multiple label spaces that correspond to different stakeholder interpretations of an appraisal category, or by providing both distributional and consensus-based scoring targets.

Negotiation also introduces expectations that can be evaluated empirically. If revisions are justified as clarifying category definitions or reducing ambiguity, one expectation is that inter-annotator reliability and abstention rates change measurably after revision. If revisions are justified as improving the match between benchmark outputs and the needs of an institution, a further expectation is that downstream uses of benchmark-selected models exhibit different error profiles. These expectations do not presume that reliability must increase; they make explicit what a revision claims to change.

Table 1 summarizes a disclosure set intended to support negotiated revision and auditing. The aim is not to standardize the content of benchmarks, but to standardize the observability of assumptions that affect what alignment scores imply.

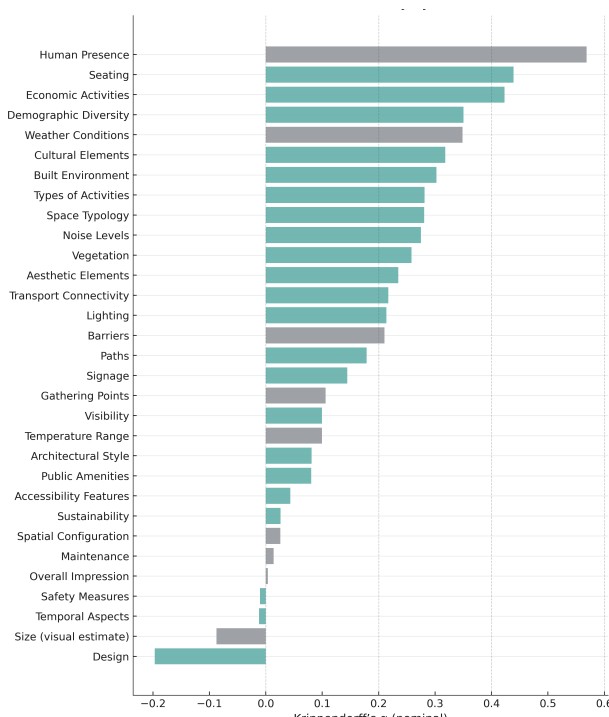

*Figure 6.* Inter-annotator reliability by dimension. Bars show Krippendorff's $\alpha$ (nominal).

Treating negotiation as part of benchmark governance raises questions about representation, authority, and resource requirements. These questions are not resolved by a single protocol. The position of this paper is that explicit procedures and versioned specifications provide a basis for contestation and revision that is not available when benchmark assumptions remain implicit.

## 10. Ethics and Privacy

The case study used human participants and street imagery. Participants gave informed consent and were compensated. Annotations are released only in aggregate, without personal data. Images were captured in public spaces, with faces and license plates blurred and verified. Synthetic images were generated using Stable Diffusion XL.

More broadly, VLM-based urban perception raises privacy and legitimacy concerns. Street imagery can enable profiling, and perception labels may inform allocation or enforcement decisions. These risks motivate evaluation practices that enable contestation, document aggregation choices, and support recourse. Reliability-aware benchmarking does not resolve these issues, but it discourages treating appraisal outputs as measurements by making disagreement and abstention explicit. The benchmark is intentionally small to support documentation, community partnership, and reliability analysis, which limits geographic coverage and statis-

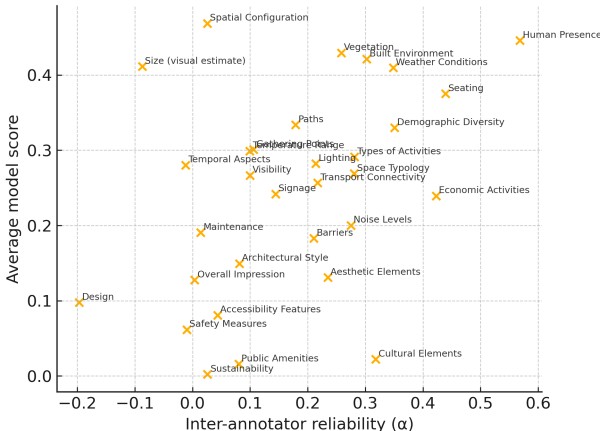

*Figure 7.* Relationship between human reliability (Krippendorff's $\alpha$) and average model score by dimension.

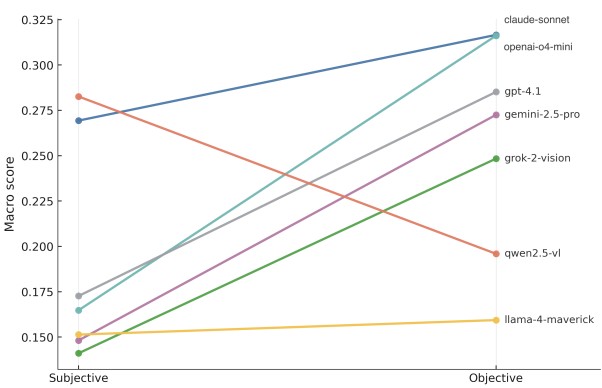

*Figure 8.* Appraisal versus observable performance by model. Each line connects a model's macro score on the appraisal subset to its macro score on the observable subset.

tical power. Annotations reflect judgments from a specific community and may not generalize. Deterministic French-to-English normalization and CSV parsing can under-score models if mapping errors occur. Evaluation is zero-shot under a fixed prompt and does not cover interaction, fine-tuning, or multi-image context. Some dimensions include non-exclusive options, making single-label scoring a coarse summary that can lower reliability. Finally, although no personal data are released, linking perception labels to places can still enable stigmatizing narratives, a risk not addressed by benchmark design alone.

## 11. Alternative Views

A first alternative view is that disagreement can be treated as annotation noise and that majority vote labels suffice for benchmarking. Under this view, reliability metadata is not central because the benchmark label is treated as a

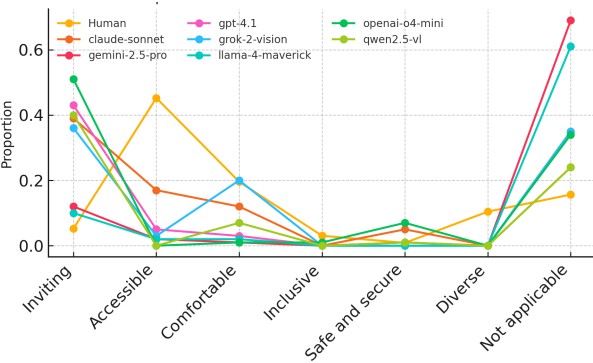

*Figure 9.* Overall Impression: distribution across annotators and models. Curves indicate the proportion of images assigned to each category.

*Table 1.* Disclosure elements for reliability-aware and negotiated benchmarks.

| Element | Content |
| --- | --- |
| Label specification | Dimension definitions, allowed values, and semantics of abstention and ties. |
| Judgment collection | Recruitment, language, interface, overlap structure, and compensation. |
| Reliability report | Per-dimension reliability measure(s), number of raters per item, and the denominator used for scoring. |
| Aggregation and scoring | Consensus rule, treatment of missingness, and scoring metrics. |
| Model interface | Prompt contract, structured output constraints, and parser rules. |
| Revision record | Version identifier, change log, rationale, and participating stakeholders. |

target. The empirical anchor provides evidence that this assumption does not hold uniformly for perception tasks that include appraisal categories. Reliability varies across dimensions, and model alignment co-varies with reliability. If the benchmark label is treated as ground truth, low scores can be read as model failure even when the target is unstable. Reliability-aware reporting changes the interpretation by distinguishing disagreement about category boundaries from disagreement about recoverable visual evidence.

A second alternative view is that benchmarks should exclude appraisal categories and focus only on dimensions closer to visible attributes. This approach can reduce ambiguity and can support stable labels. The limitation is that appraisal categories are part of how urban perception tools are proposed and used, including in participatory settings where stakeholders request summaries of comfort, accessibility, or safety. Excluding these categories shifts the problem from evaluation to deployment: models may still be used to generate appraisals, but without benchmark evidence about how these outputs relate to human judgments or how abstentions

are handled. A reliability-aware benchmark can instead treat appraisal categories as pluralistic targets and can report disagreement and distributional mismatch as outcomes rather than as errors to be eliminated.

A third alternative view is that negotiation and co-production are not feasible for machine learning benchmarks because they reduce scalability and comparability. In this view, benchmarks should prioritize standardization to support model ranking and progress tracking. Negotiation and standardization can coexist. A benchmark can provide a standardized evaluation harness while documenting who defined the label space, how disagreement was handled, and what alternative label spaces were considered. If a negotiated revision yields a different label space, the resulting benchmark can be treated as a different task specification rather than as a revision that invalidates prior results. Lifecycle structures for such revisions have been proposed in co-produced urban AI (Mushkani et al., 2025a).

A fourth alternative view is that disagreement can be avoided by replacing human targets with automated evaluation, for example by using models as judges for appraisal outputs. This approach can produce internally consistent scores, but it shifts the specification problem rather than resolving it. If the judge embeds a fixed interpretation of appraisal categories, then the evaluation hides plurality rather than representing it. For governance-relevant targets, this reduction in observability can limit contestation and can reduce the ability of affected groups to challenge category definitions or abstention semantics.

## 12. Conclusion

Benchmarking VLMs for urban perception depends on design choices such as categories, aggregation, translation, and abstention that shape what scores mean. The Montreal benchmark shows that alignment varies by dimension, co-varies with human reliability, and diverges from annotators in abstention behavior on appraisal categories. These findings suggest treating disagreement and abstention as outcomes, reporting reliability alongside alignment, and viewing benchmark specifications as revisable when used in governance.

A negotiation-oriented view of benchmarking connects evaluation to procedures for revising labels, prompts, and scoring in response to contestation or context change. Work in participatory urban AI, including pluralistic datasets, co-production lifecycles, collective prompting, and negotiative alignment, offers such procedures. For machine learning, this implies that benchmark quality for appraisal targets depends not only on statistical design but also on transparent assumptions and revision mechanisms.

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

# A. Call to Action

For benchmark creators, reliability-aware reporting can be treated as a documentation requirement for appraisal targets. This includes reporting inter-annotator reliability per dimension, reporting abstention rates, and disclosing how abstentions enter aggregation and scoring. Where feasible, benchmarks can release distributions of judgments in addition to consensus targets, since distributional information can reveal mismatch even when pointwise agreement is limited. For multilingual annotation, normalization rules and translation mappings should be documented and made reproducible.

For model developers and platform providers, evaluation can be supported by interfaces that represent abstention and uncertainty explicitly and by logs that record model versioning and prompting conditions. When models are offered through hosted APIs, version drift can change benchmark results; disclosure reduces ambiguity about what was evaluated. Tooling that supports constrained structured outputs reduces evaluation variance introduced by parsing and makes benchmark artifacts easier to audit.

For conference organizers and reviewers, a reporting norm can be introduced for papers that claim perceptual capability on urban imagery or other perception tasks with appraisal labels. This norm can require a description of how labels were constructed, whether disagreement and abstention were present, and how they were treated in scoring. Since many benchmarks already collect multiple annotations, this norm primarily concerns reporting and interpretation.

For urban institutions and civil society organizations, VLM outputs can be treated as inputs to deliberation rather than as measurements. When perception labels are used, stakeholders can request reliability reports, disagreement distributions, and documentation of the label definitions and scoring policy. When stakeholders contest a label space, negotiated revisions can be documented and re-evaluated rather than treated as out-of-scope.

# B. Perception Grid

The benchmark uses **30** dimensions.

*Table 2.* Perception grid: dimensions, allowed values, and type.

| Dimension | Allowed Values | Type |
|---|---|---|
| Space Typology | Park; Street; Square; Courtyard; Garden; Waterfront; Public plaza; Alley; Playground; Not applicable | multiple |
| Spatial Configuration | Open; Enclosed; Semi-enclosed; Structured; Organic; Not applicable | single |
| Size (visual estimate) | Small (<500 m$^2$); Medium (500-2000 m$^2$); Large (>2000 m$^2$); Not applicable | single |
| Lighting | Natural lighting; Artificial lighting; Well lit; Poorly lit; Shaded areas; Not applicable | multiple |
| Maintenance | Clean; Dirty; Well maintained; Neglected; Recently renovated; Not applicable | single |
| Vegetation | Trees present; Too much greenery; Little greenery; Grass present; Bushes present; Flower beds present; No vegetation; Not applicable | multiple |
| Paths | Paved paths present; Unpaved paths present; Wide paths present; Narrow paths present; Linear paths present; Curved paths present; Intersecting paths present; Dead-end paths present; Not applicable | multiple |
| Seating | Benches present; Chairs present; Picnic tables present; Custom seats present; Movable seats present; No seating; Not applicable | multiple |
| Built Environment | Modern buildings present; Historic buildings present; Residential buildings present; Commercial buildings present; Mixed-use buildings present; Vacant lots present; Not applicable | multiple |
| Signage | Informational signs present; Decorative signs present; Directional signs present; Interactive signs present; No signage; Not applicable | multiple |

| Dimension | Allowed Values | Type |
|---|---|---|
| Human Presence | Crowded (>50 people); Moderately populated (20-50 people); Sparsely populated (<20 people); Empty; Not applicable | single |
| Types of Activities | Recreational activities present; Leisure activities present; Commercial activities present; Transportation activities present; Cultural activities present; Social activities present; Sports activities present; Religious activities present; Not applicable | multiple |
| Accessibility Features | Ramps present; Handrails present; Tactile paving present; Elevators present; Wide entrances present; Accessible restrooms present; No accessibility features; Not applicable | multiple |
| Visibility | Clear sight lines; Obstructed views present; Panoramic views present; Hidden corners present; Not applicable | multiple |
| Safety Measures | Surveillance cameras present; Security personnel present; Safety lighting; Emergency exits present; Safety signs present; Fences present; Walls present; Not applicable | multiple |
| Barriers | Physical barriers present (fences, walls); Natural barriers present (rivers, hills); No barriers; Not applicable | single |
| Aesthetic Elements | Bright colours present; Dark colours present; Monochrome elements present; Murals present; Sculptures present; Street art present; Water features present; No decorative elements; Not applicable | multiple |
| Architectural Style | Traditional buildings present; Contemporary buildings present; Eclectic buildings present; Vernacular buildings present; Post-modern buildings present; Brutalist buildings present; Not applicable | multiple |
| Gathering Points | Central gathering point present; Edge gathering points present; Gathering points near monuments present; Informal gathering points present; No gathering points; Not applicable | single |
| Demographic Diversity | Variety in group sizes; Presence of family groups; Presence of mixed-age groups; Presence of mobility aids; Not applicable | multiple |
| Design | Wheelchair-accessible features present; Braille signage present; Multilingual signs present; Gender-neutral restrooms present; Adapted play equipment present; No design features; Not applicable | multiple |
| Weather Conditions | Sunny; Rainy; Snowy; Cloudy; Windy; Foggy; Not applicable | single |
| Temperature Range | Hot (>30 °C); Warm (20-30 °C); Cool (10-20 °C); Cold (<10 °C); Not applicable | single |
| Noise Levels | Quiet; Moderate; Loud; Traffic noise present; Construction noise present; Natural sounds present; Not applicable | multiple |
| Temporal Aspects | Daytime; Night; Weekday; Weekend; Seasonal variations; Not applicable | single |
| Public Amenities | Restrooms present; Water fountains present; Information kiosks present; Trash bins present; Play areas present; Fitness equipment present; Not applicable | multiple |
| Economic Activities | Street vendors present; Markets present; Shops present; Cafés present; No commercial activities; Not applicable | multiple |
| Transport Connectivity | Public transport access present; Bicycle lanes present; Pedestrian paths present; Parking spaces present; Carpool points present; Not applicable | multiple |
| Cultural Elements | Historic monuments present; Monuments present; Culturally significant features present; Public art installations present; Not applicable | multiple |

| Dimension | Allowed Values | Type |
|---|---|---|
| Sustainability | `Recycling bins present; Green building features present;`
`Use of renewable energy present (e.g., solar panels);`
`Water conservation measures present; Not applicable` | multiple |
| Overall Impression | `Inviting; Accessible; Comfortable; Inclusive; Safe and`
`secure; Diverse; Cannot judge; Not applicable` | single |

**Tokenization note.** The harness expects a single CSV line per image with 30 comma-separated fields, one per dimension above; multi-label selections are joined with semicolons and no spaces (e.g., `Natural lighting;Well lit`). The output CSV written to disk adds two columns around this line-level response: a leading `Image_ID` and a trailing `Comments` field used for parser diagnostics. The model never outputs `Image_ID`.

## C. Image Source Effects (real vs. synthetic)

Panels $p1$–$p5$ consist of photorealistic synthetic scenes and $p6$–$p10$ are real photographs (50 images each). Figure 10 breaks down agreement by source. Agreement on synthetic images is lower than on photographs across models in this benchmark, while the ranking by model does not change.

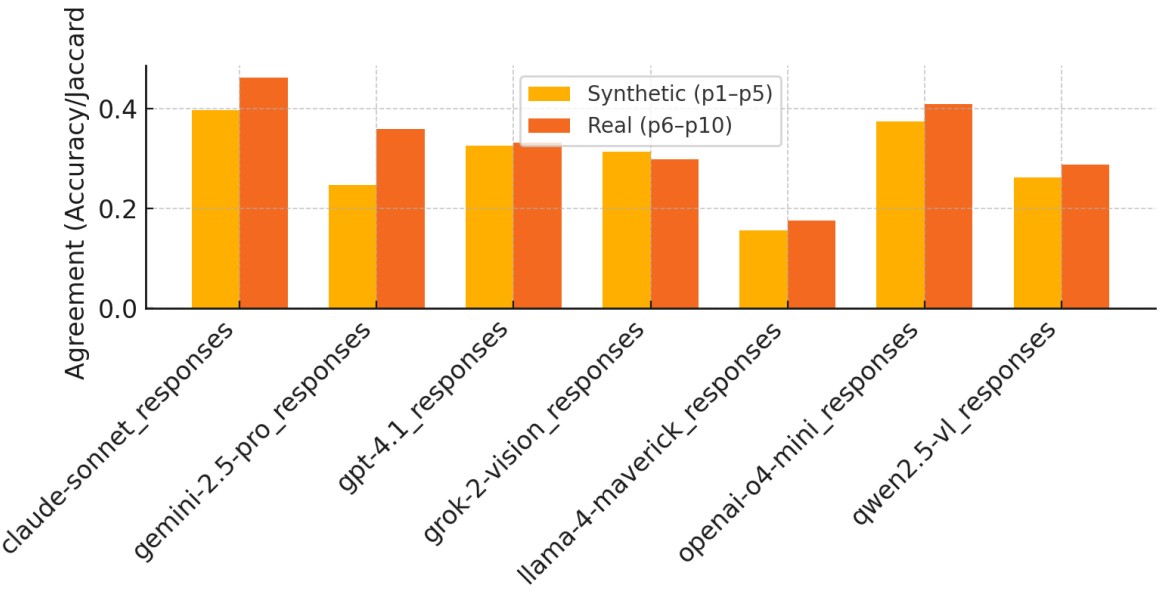

*Figure 10.* Model agreement by image source. Real photos ($p6$–$p10$) and synthetic renders ($p1$–$p5$).

## D. Prompt and Parsing Details

The inference script provides a deterministic system prompt that lists the 30 columns in the order shown in Table 2, requires only a single CSV line with 30 fields and no header or commentary, instructs models to use `Not applicable` when unclear, and specifies the semicolon rule for multi-label items. Parsing is rule-based: replies are split on commas (with a CSV fallback), trimmed and mapped to codebook tokens, and padded or truncated to the expected length. Any non-conforming reply is captured in `Comments` and excluded from scoring.

## E. Implementation Notes

To reduce ambiguity about image access and to improve run-to-run consistency, the benchmark harness embeds local images as base-64 data URIs in the chat message (no remote fetches), sets `temperature=0` and `top_p=1` with fixed token limits while logging model version strings and run timestamps, uses retry and backoff for transient API errors while logging raw replies for diagnostics, discovers images under `p1`–`p10` panel folders (100 files), and enforces the one-line

CSV contract with a deterministic post-processor that writes `Image_ID + 30 fields + Comments` to disk. The harness also exposes an alternative treatment of `Not applicable` that counts it as a regular label.

## F. Prompt Template

```
Prompt contract (model-facing)

def build_system_prompt() -> str:
    lines = [
        "You are an urban-perception assessor.",
        "Return ONLY a single CSV line (no header) with 30 comma-separated fields,",
        "one field per dimension, in the exact order listed below.",
        "For multi-select dimensions, join choices with semicolons (no spaces).",
        "If evidence is unclear or absent, output Not applicable.",
        "Do NOT include an image id, quotes, or commentary; just the CSV line.",
        "",
        "Example (for the Lighting dimension only, which is column 4):",
        ",,,Natural lighting;Well lit,,,,,,,,,,,,,,,,,,,,,,,,,,,",
        "",
        "Column order and allowed values:",
    ]
    for i, dim in enumerate(GRID, 1):
        flag = " (multiple)" if dim.multiple else " (single)"
        lines.append(f"{i}. {dim.name}{flag}: " + "; ".join(dim.variables))
    lines.append("Return just the CSV line, nothing else.")
    return "\n".join(lines)
```

