# OpenReview forum: "Position: Benchmarks for Vision–Language Models in Urban Perception Should Be Reliability-Aware and Negotiated"
_ICML.cc/2026/Position_Paper_Track — ICML 2026 Position Paper Track regular_

### Official Review · Reviewer_HjJu · 2026-02-17

**Significance:** 3
**Argument Clarity:** 3
**Rating:** 4
**Confidence:** 1

**Questions:**

How was the observable/appraisal partition defined? By author judgment or some empirical criterion (e.g., $\alpha$ threshold)?

**Alternative Views Section:**

Yes

**Compliance With Llm Reviewing Policy A Conservative:**

Affirmed.

**Discussion Potential:**

3

**Final Justification:**

The rebuttal adequately addressed my concerns regarding the negotiation procedure, the granularity of Figure 7, and the observable/appraisal partition. I raise my score.

**Paper Summary:**

This paper argues that: VLM urban perception benchmarks should report inter-annotator reliability. Moreover, they should treat disagreement as informative and make label spaces negotiable. The authors show that VLMs have agreement that covaries with human reliability using a Montreal benchmark that contains 100 images with 12 annotators.

**Position:**

Yes

**Position In Title:**

Yes

**Related Work:**

2

**Strengths And Weaknesses:**

Strengths:
- The benchmark design is detailed and careful.
- The correlation between human reliability and model agreement in Figure 7 supports the paper's main claim. Specifically, reporting scores without reliability contexts could be misleading.

Weaknesses:
- The argument of reporting reliability is sound. However, the negotiation procedure is somewhat vague and lacks a working example. For example, who initiates revision and what triggers re-annotation?
- The reliability alignment correlation in Figure 7 has only 30 data points.

**Support:**

3

---

> ### Author Rebuttal · Authors · 2026-03-29
>
> We thank the reviewer for the thoughtful questions. We address the three points directly.
>
> 1 - On the negotiation procedure. The paper’s intent is not to prescribe one universal governance actor, because who has authority to revise a benchmark depends on the institution and deployment context. What the paper does specify is the object and mechanics of revision. Section “Benchmark negotiation and versioning” defines the benchmark specification as including the label ontology, abstention semantics, language/normalization mappings, and prompt contract. It also defines comparability through versioning: if categories or scoring policies are revised, results should be reported across versions rather than folded into a single context-free benchmark. The paper further makes negotiation empirically accountable: revisions that claim to clarify category definitions or policy semantics should induce measurable changes in reliability, abstention rates, and possibly downstream error profiles after re-annotation. A concrete trigger is therefore contestation of category meaning, abstention semantics, or intended decision use; re-annotation is warranted when those specification elements change. For example, the current schema already separates Accessibility Features from the appraisal label Accessible in Overall Impression. If stakeholders contest whether “Accessible” can be judged from a static image alone, that is precisely the kind of specification dispute the paper argues should become an explicit benchmark revision rather than an implicit assumption.
>
> 2 - On Figure 7 having 30 points. That is correct, because the unit of analysis is the dimension. This is intentional: the paper’s claim is that measurement properties are dimension-specific, so each point summarizes how one dimension behaves across the overlapping annotations and model outputs. In other words, the evidence base is not “30 observations” in the ordinary sense; each point aggregates scene-level judgments over the benchmark. We do not present the figure as a universal law or as the only evidence for the position. It is one of three convergent analyses, alongside the observable-versus-appraisal gap and the Overall Impression distributional mismatch. Together they show that score interpretation depends on the reliability and semantics of the target.
>
> 3 - On the observable/appraisal partition. This partition was defined by author judgment, using a simple criterion: whether the dimension can be answered primarily from directly visible scene evidence, or whether it requires evaluative/experiential interpretation beyond what is plainly visible. It is therefore a coarse descriptive stratification, not a learned threshold and not the main empirical claim. The stronger result is the continuous reliability analysis over all 30 dimensions in Figure 7, which does not depend on any binary partition. We agree that the criterion and subset membership should be made explicit for readers, and we will clarify this in the revision. Representative “closer to observable” dimensions are items such as Vegetation, Seating, and Human Presence; the appraisal side is centered on dimensions such as Overall Impression where category use clearly depends on interpretation.
>
> We hope this clarifies why the current paper already supports acceptance. The contribution is not a large-sample correlation claim; it is a benchmark-design argument made concrete by a fully specified case study that exposes how disagreement, abstention, and versioned task specifications change what a VLM score means.

---

> > ### Author Rebuttal · Reviewer_HjJu · 2026-04-03
> >
> > All three concerns are adequately addressed by the rebuttal. I am willing to raise my score.

---

### Official Review · Reviewer_VTrG · 2026-03-05

**Significance:** 3
**Argument Clarity:** 3
**Rating:** 5
**Confidence:** 4

**Questions:**

This paper has good breadth and completeness of position discussion. My main concern is the depth of the author's position analysis and discussion. Please address the weaknesses mentioned above.

**Alternative Views Section:**

Yes

**Compliance With Llm Reviewing Policy A Conservative:**

Affirmed.

**Discussion Potential:**

3

**Final Justification:**

My concerns have been fully addressed by the author. I maintain my original rating, recommend accepting this paper.

**Paper Summary:**

This paper argues that evaluating VLMs for urban perception requires more careful benchmarking because many labels (e.g., comfort, safety, or accessibility) are subjective and often involve disagreement among people. Using a small benchmark of 100 street scenes from Montreal annotated across 30 dimensions, the paper evaluates some VLMs and show that model performance tends to track how consistently humans agree on a dimension. Models perform better on clearly observable attributes than on subjective judgments. Based on these findings, the paper advocates reliability-aware benchmarks, where evaluation reports include annotator reliability, abstention behavior, and transparent documentation of how labels and scoring rules are defined.

**Position:**

Yes

**Position In Title:**

Yes

**Related Work:**

3

**Strengths And Weaknesses:**

This paper investigates a machine learning topic that appears worthy of discussion. The authors' position is well-motivated, and they employ numerous examples, discussions, analyses, and results to support their arguments. Overall, I believe the paper is relatively complete and persuasive. However, I think the paper has the following weaknesses:

1) Although the paper uses small-scale experiments to support its arguments, the dataset is too limited in size and scope, potentially making the supporting conclusions somewhat weak.
2) Whether the existing benchmark has partially solved the problem could be discussed more deeply in related work.
3) The quality of Figures 3, 4, 6, and 8 could be improved, especially the font size, which needs to be enlarged.

**Support:**

4

---

> ### Author Rebuttal · Authors · 2026-03-29
>
> We thank the reviewer for the encouraging assessment and for identifying the few places where the paper can be made sharper.
>
> On scale and scope, we agree the empirical anchor is intentionally limited. The paper states that it is not a representative sample of urban perception and not a substitute for larger datasets. Its purpose is narrower and, we would argue, already achieved: to make a benchmark-design problem observable under a fully specified protocol. Even with 100 scenes, the study contains 6,900 dimension-level human judgments across 30 dimensions and 21,000 model outputs across 7 VLMs. This is enough to show three nontrivial patterns that support the position: higher model alignment on dimensions closer to visible attributes than on appraisal dimensions, a positive association between human reliability and model agreement across dimensions, and a model-human distributional mismatch on Overall Impression, especially around abstention semantics. Larger benchmarks would improve estimate precision, but they are not required for the methodological point the paper makes.
>
> On the depth of the position analysis, our goal is not only to say “report reliability,” but to show how reliability changes the interpretation of a model score. The paper argues that, for pluralistic targets, alignment must be read as conditional on measurement properties of the target. It then operationalizes this in a concrete way through per-dimension reliability, abstention handling, distributional reporting, and a versioned benchmark-specification view. In that sense, the position is already more than a call for better documentation: it is a claim about what counts as valid model comparison when the target itself is contested or unstable.
>
> We also agree that prior work which partially solves adjacent pieces could be contrasted more explicitly. Our view is that earlier benchmarks and evaluation frameworks each contribute an important component, but not the combination studied here. For example, Place Pulse captures crowd judgments of urban scenes, CheXpert treats uncertainty explicitly in expert medical labeling, toxicity work shows that annotator identities and beliefs matter, and VLM evaluation toolkits standardize harnesses. What is new here is the synthesis for VLM urban perception: disagreement and abstention are treated as first-class measurement outcomes, and benchmark categories/scoring policies are treated as revisable task specifications rather than fixed ground truth.
>
> Finally, thank you for flagging figure readability. We agree that some figures can be made easier to read, especially the ones you noted, and we will enlarge font sizes and improve layout in the revision.
>
> We appreciate the positive overall reading. Our hope is that the paper is evaluated as a methodologically important position with a concrete empirical anchor: it does not merely argue that subjective benchmarks are hard, but shows exactly how current VLM evaluation can become misleading when reliability and abstention remain hidden.

---

> > ### Author Rebuttal · Reviewer_VTrG · 2026-04-01
> >
> > Thank you to the author for the detailed rebuttal. My concerns have been fully addressed by the author.
> >
> > Therefore, I maintain my original rating, recommend accepting this paper at this stage.

---

### Official Review · Reviewer_JCzw · 2026-03-12

**Significance:** 2
**Argument Clarity:** 2
**Rating:** 4
**Confidence:** 3

**Questions:**

Please see above weaknesses.

**Alternative Views Section:**

Yes

**Compliance With Llm Reviewing Policy A Conservative:**

Affirmed.

**Discussion Potential:**

2

**Final Justification:**

I raised the score to borderline accept.

Justification is below:
The rebuttal addresses how the position links to detailed ML problems and how the formulation might look like. Also, it replied to the data scale concern, and visualization concern.

**Paper Summary:**

This paper focuses on the benchmarks for vision-language model in urban perception. The position states that benchmarking VLMs for urban perception should treat disagreement and abstention as measurement outcomes, report inter-annotator reliability with model alignment, and treat label space and scoring policy as negotiable artifacts.

In order to demonstrate the position, this paper collects a small dataset of 100 scenes and recruits 12 annotators as an empirical anchor. The various VLM models performance and annotators' performance are reported and comparied either as a whole or dimension-wise.

Discussions are provided and alternative views as well as call to action are also provided.

**Position:**

Yes

**Position In Title:**

Yes

**Related Work:**

2

**Strengths And Weaknesses:**

### Strengths
- This position paper focuses on the VLM application to urban perception problem, with unique properties such as common disagreement / abstention for labels.
- A small dataset is collected as an empirical anchor and 12 annotators are recruited for rating.
- Various VLMs' results are compared with annotators for demonstrating the position.
- A series of results/figures are reported and discussions are provided.

### Weaknesses
- This position paper focuses on a special VLM application problem: urban perception, which covers the intersection of AI and urban/govermance science. However, the paper writing and language are mainly from the descriptive and application perspective, deviating from the ML common practices. In ML community, problem is usually defined with proper symbols and formulations. This paper is more of social science or urban science style. Threfore its scope and potential benefits for ML community are not significant.
- The empirical anchor only contains 100 scenes, which is fairly small and the results is less convincing or illustrative.
- No visualization of real dataset images or real labels are shown to elaborate the problem of disagreement or negotiation.
- This paper simply compares various VLM results with human raters. It is unclear what the potential benefits or discussion viewpoints can be made in the ML community.
- The related work is too short and few.

**Support:**

2

---

> ### Author Rebuttal · Authors · 2026-03-29
>
> We thank the reviewer for the candid assessment. We believe the main disagreement is about the paper’s contribution type. This is not “a small urban application paper”; it is an ML evaluation paper about what a benchmark means when the target labels are pluralistic, abstentions are meaningful, and outputs may inform governance decisions.
>
> A compact way to read our formulation is that a benchmark for these tasks is not only (images, labels). It is also a label ontology, an annotator distribution, an aggregation rule, abstention semantics, a scoring rule, and a prompt/parser contract. On subjective targets, omitting those terms under-specifies the task. The paper’s core contribution is to make those hidden benchmark variables explicit and auditable. That is squarely an ML concern, because benchmark design determines what “progress” means and what kinds of systems are optimized. In that sense, the contribution is analogous in spirit to how uncertainty labels changed evaluation in medical imaging and how annotator-conditioned labels changed evaluation in toxicity detection; the paper brings that insight to VLM benchmarking for urban perception.
>
> For this reason, the paper intentionally uses accessible prose rather than introducing heavy notation for its own sake. Still, the work is already operationalized in standard ML terms: fixed label sets for 30 dimensions, explicit single-choice vs multi-label scoring, Krippendorff’s alpha per dimension, accuracy/Jaccard alignment metrics, deterministic French-to-English normalization, a fixed zero-shot prompt contract, and a deterministic parser/harness across 7 VLMs. In other words, the methodological object is precise even though the writing aims to bridge ML, urban science, and governance audiences.
>
> Regarding scale, we agree the benchmark is not large in number of scenes, and the paper says so explicitly. But the empirical anchor is richer than “100 images” suggests: 100 scenes x 30 dimensions x 1-3 raters yields 6,900 dimension-level human judgments, and 7 VLMs yield 21,000 model outputs under the same contract. The paper does not present these as population-level urban perception estimates or as a definitive leaderboard. It uses them to demonstrate three concrete evaluation phenomena that already matter for ML benchmarking: model agreement tracks human reliability across dimensions; models perform better on dimensions closer to visible attributes than on appraisal dimensions; and on Overall Impression, models and humans differ in abstention/distributional behavior even when a point score alone would hide that mismatch.
>
> On the absence of qualitative image examples, we agree they could aid intuition. We prioritized benchmark-level evidence over anecdotal cases because the paper’s claim is about evaluation semantics, not about one or two illustrative scenes. The quantitative figures make disagreement and abstention visible at the benchmark level, which is the right evidentiary level for the position. In the revised version, we will also add a compact qualitative panel with representative scenes and per-image label distributions.
>
> On benefits to the ML community, we see four concrete ones already in the paper: (1) reliability as required metadata for interpreting scores on pluralistic targets, (2) abstention as measurable model behavior rather than discarded noise, (3) benchmark negotiation/versioning as a way to preserve comparability while allowing contested task specifications to be revised, and (4) disclosure of prompt/parser contracts as part of the benchmark itself. These are not urban-only ideas; urban perception is the motivating stress test because it combines visible structure with governance-laden appraisal.
>
> Finally, on related work: the paper already connects urban perception datasets, VLM evaluation, geographic/spatial reasoning, pluralistic labeling in toxicity detection, uncertainty labeling in medical imaging, and dataset documentation. We agree that the distinction from prior benchmarks that partially address adjacent pieces can be made even more explicit. The key distinction is that prior work typically contributes either a dataset, or an uncertainty label scheme, or an evaluation harness. Our paper contributes the benchmark-governance principle that ties reliability, disagreement, abstention, and versioned task specification together for VLM evaluation.
>
> We hope this clarifies the intended contribution and why it is significant for ML, not only for urban science.

---

> > ### Author Rebuttal · Reviewer_JCzw · 2026-04-03
> >
> > The rebuttal addresses how the position links to detailed ML problems and how the formulation might look like. Also, it replied to the data scale concern, and visualization concern.

---

### Official Review · Reviewer_Mzko · 2026-03-12

**Significance:** 2
**Argument Clarity:** 3
**Rating:** 4
**Confidence:** 2

**Questions:**

See above

**Alternative Views Section:**

Yes

**Compliance With Llm Reviewing Policy A Conservative:**

Affirmed.

**Discussion Potential:**

2

**Final Justification:**

Thanks for the rebuttal. While i am not an expert in related fields and not quite familar with related works, i lead slightly towards positive opinions and keep my score.

**Paper Summary:**

This paper presents a position on how benchmarks for vision–language models (VLMs) should be designed when applied to urban perception tasks. The submission analyzes an important concept: the reliability and interpretability of benchmarks when the target labels are subjective, pluralistic, and potentially contested. The authors argue that conventional benchmarking practices such as collapsing multiple annotations into a single ground-truth label are insufficient for tasks involving human appraisal categories like comfort, safety, and accessibility. Instead, they propose reliability-aware benchmarking that explicitly reports inter-annotator agreement, disagreement, and abstention behaviors as evaluation signals.

**Position:**

Yes

**Position In Title:**

Yes

**Related Work:**

3

**Strengths And Weaknesses:**

I want to first admit that i am not a researcher in related fields, and i just want to give the review from a general researcher.

Strengths:

1. Timely and important research question. The paper addresses an increasingly relevant issue as vision–language models are applied to real-world urban analysis and planning tasks. It highlights a critical mismatch between traditional ML benchmarking assumptions and subjective human perception tasks.

2. Clear conceptual contribution on reliability-aware benchmarking. The paper provides a well-articulated argument for treating disagreement and abstention as meaningful outcomes rather than noise. The conceptual framework connects ideas from ML evaluation, human annotation reliability, and governance contexts.

3. Empirical evidence supporting the position. Although primarily a position paper, the authors include an empirical case study demonstrating the relationship between human reliability and model performance. The results showing better performance on observable attributes versus appraisal categories provide useful insights into VLM limitations.

4. Thoughtful evaluation design and reproducibility considerations. The benchmark includes detailed annotation protocols, deterministic prompting, and transparent scoring policies. The paper also documents prompt contracts, parsing rules, and abstention handling, which improves reproducibility.

Weaknesses:

1. Limited scale of the empirical benchmark. The dataset contains only 100 images and 12 annotators, which restricts statistical power and generalizability.

2. Empirical evaluation is relatively shallow. The experiments primarily report descriptive statistics and correlations rather than deeper analysis of model behavior. For example, the paper could further analyze which visual cues or dimensions cause disagreement between models and annotators.

3. Ambiguity in the operationalization of “negotiated benchmarks”. While the paper advocates benchmark negotiation and versioning, the practical implementation of these processes is not fully specified. Questions remain about how stakeholders are selected, how disagreements are resolved, and how comparability across versions is maintained.

4. Limited comparison with alternative evaluation frameworks. The paper argues against majority-vote ground truth but does not experimentally compare multiple aggregation methods or uncertainty-aware evaluation metrics. Including such comparisons could strengthen the empirical support for the proposed approach.

Actually, i lean towards borderline. But considering the potential effects, i give a borderline accept rating.

**Support:**

3

---

> ### Author Rebuttal · Authors · 2026-03-29
>
> We thank the reviewer for the careful reading and the constructive framing. A central clarification is that this is a position paper about benchmark semantics, not a population-generalization paper. The empirical anchor is intentionally scoped to expose a concrete failure mode in current VLM evaluation: when pluralistic targets are collapsed to a single label, the reported score conflates model capability with target reliability and abstention semantics.
>
> On scale: we agree the benchmark is small in number of scenes, and the paper states this explicitly. But it is not anecdotal in number of judgment events: 100 scenes x 30 dimensions x 1-3 raters yields 6,900 dimension-level human judgments, and the fixed zero-shot evaluation adds 21,000 model-dimension outputs across 7 VLMs. This is sufficient for the paper’s purpose, which is not to rank models with high external-validity claims, but to show that benchmark interpretation changes once reliability and abstention are made visible.
>
> On depth of analysis: we intentionally emphasize descriptive analyses because the paper’s claim is about what evaluation reports should contain and how scores should be interpreted. Even so, the current results already do more than report macro scores. They show: (1) a systematic observable-versus-appraisal gap across models, (2) a positive relationship between dimension-level human reliability and dimension-level model agreement, and (3) distributional mismatch on Overall Impression, including different uses of Not applicable/Cannot judge. These analyses already distinguish cases where low score is consistent with missing visual evidence from cases where it is consistent with unstable target semantics or stronger model priors. Cue-level attribution would be valuable follow-on work, but it is not required to establish the benchmark-design claim.
>
> On “negotiated benchmarks”: the paper does specify the operational object of negotiation. Section “Benchmark negotiation and versioning” defines the benchmark specification as including the label ontology, abstention semantics, language/normalization mappings, and prompt contract. It also specifies comparability through versioning, not by pretending one label space is context-free. The paper further makes the process empirically accountable: if a revision is meant to clarify a category or change its policy semantics, one should observe shifts in reliability, abstention rates, and possibly downstream error profiles after re-annotation. We deliberately do not hard-code a single universal stakeholder protocol, because authority and representation depend on the institutional setting; the contribution is to make the revisable artifact and the expected evidence of revision explicit.
>
> On alternative aggregation/uncertainty-aware metrics: our aim is not to argue that one replacement metric should dominate majority vote. It is to show why majority-vote scores in isolation are insufficient. We therefore score against the conventional consensus target precisely to speak to current practice, then show why reliability, disagreement, and abstention must accompany that score. The appendix already exposes one policy sensitivity check by allowing Not applicable to be treated either as non-response or as an ordinary label.
>
> We hope this clarifies why the current paper already meets the acceptance bar: it contributes a general evaluation principle, a concrete disclosure framework, and a fully specified empirical anchor that makes the principle observable.

---

> > ### Author Rebuttal · Reviewer_Mzko · 2026-04-02
> >
> > Thanks for the rebuttal. While i am not an expert in related fields, i lead slightly towards positive opinions and keep my score.

---

### Decision · Program_Chairs · 2026-04-30

**Decision:**

Accept (regular)

**Comment:**

Overall, this paper received sufficient support for acceptance. Reviewers emphasized the timeliness and importance of the topic, as well as its potential impact on the community.